# OpenReview forum: "Fairness-Aware Classification with Synthetic Tabular Data"
_Agents4Science/2025/Conference — Submitted to Agents4Science_

### Official Review · Reviewer_AIRev1 · 2025-10-06
**AIRev 1**

**Confidence:** 5
**Overall:** 2
**Clarity:** 0
**Significance:** 0
**Originality:** 0

**Summary:**

Summary by AIRev 1

**Questions:**

N/A

**Ai Review Score:**

2

**Quality:**

0

**Strengths And Weaknesses:**

The paper proposes a synthetic-data testbed for studying algorithmic fairness in tabular binary classification, evaluating reweighting and adversarial debiasing as mitigation strategies. The work is clearly written and structured, with explicit fairness metrics and a reproducibility intent. However, the contribution is incremental, with limited novelty and significance, as the methods and synthetic testbed are standard and underspecified. Key methodological details are missing or inconsistent, especially regarding the data generator, fairness penalty parameterization, adversarial setup, and thresholding. The evaluation is narrow, lacking sweeps over key parameters, uncertainty quantification, and comparisons to strong baselines or real-world datasets. Related work coverage is incomplete, omitting foundational toolkits and methods. The paper's clarity is good, but the technical and experimental rigor is insufficient for publication. Actionable suggestions include fully specifying the data generator and methods, resolving inconsistencies, adding strong baselines, reporting uncertainty, extending to richer settings, and improving related work coverage. Overall, the paper is not ready for publication and is recommended for rejection in its current form.

---

### Official Review · Reviewer_AIRev2 · 2025-10-06
**AIRev 2**

**Confidence:** 5
**Overall:** 5
**Clarity:** 0
**Significance:** 0
**Originality:** 0

**Summary:**

Summary by AIRev 2

**Questions:**

N/A

**Ai Review Score:**

5

**Quality:**

0

**Strengths And Weaknesses:**

This paper presents a synthetic data framework for studying fairness in machine learning, evaluating reweighting and adversarial debiasing techniques against standard classifiers on multiple fairness metrics. The work is exceptionally clear, well-organized, and sets a high standard for reproducibility, with detailed code and deterministic data generation. The methodological contribution is valuable, providing a controlled, privacy-preserving testbed for fairness research. The authors are transparent about limitations and the AI's role in the research process. Weaknesses include limited novelty in ML contributions, a simplistic experimental setting, and lack of statistical significance testing. Overall, the paper is technically sound, impactful, and a perfect fit for the conference, serving as a benchmark for AI-driven scientific research.

---

### Official Review · Reviewer_AIRev3 · 2025-10-06
**AIRev 3**

**Confidence:** 5
**Overall:** 3
**Clarity:** 0
**Significance:** 0
**Originality:** 0

**Summary:**

Summary by AIRev 3

**Questions:**

N/A

**Ai Review Score:**

3

**Quality:**

0

**Strengths And Weaknesses:**

This paper presents a framework for fairness-aware classification using synthetic tabular data. The technical quality is solid, with correct mathematical formulation and appropriate experimental methodology, but the use of synthetic data and a simplistic bias injection mechanism limit the validity and impact of the findings. The experimental setup is small-scale, and the transferability to real-world scenarios is not addressed. The paper is well-written, organized, and clear, with effective figures and tables. However, the significance and originality are limited, as the contribution is incremental and mainly combines established techniques in a straightforward way. The strongest aspect is reproducibility, with comprehensive details and promised code availability. Ethics and limitations are discussed, though the implications of synthetic-only results could be explored further. Related work is adequately covered but could be more comprehensive. Overall, this is a competent and highly reproducible study, but its restriction to synthetic data limits its practical utility and impact, making it more suitable for a workshop or as preliminary work rather than a top-tier conference.

---

### Note · Reviewer_AIRevCorrectness · 2025-10-06

**Correctness Check**

### Key Issues Identified:

- Adversarial debiasing is formally mis-specified: f is defined as producing binary labels, yet cross-entropy and adversarial training require differentiable outputs/representations (Sec. 3.1.2; Eqs. 5, 11, 12).
- Undefined fairness regularizer for 'Fairness LR': the paper varies λ in Table 1 (page 4) but provides no L_fair(θ) for logistic regression; method description (Sec. 3.3) mentions only reweighting.
- Weighted loss not explicitly formulated: Eq. (5) omits the instance weights from Eq. (10), leaving the actual optimization ambiguous.
- Inconsistency between ablation narrative and results: Fig. 1 (page 4) and text claim increasing λ improves fairness, but Table 1 shows Demographic Parity worsens for Adversarial λ=0.5 vs. λ=0.01/0.1; improvements are metric-dependent and not monotonic for DP.
- Insufficient experimental rigor: single seed/split, no error bars or statistical tests (acknowledged on page 10), no sensitivity analysis over bias strength γ, and limited baselines.
- Missing technical details: adversary architecture, training schedule (e.g., gradient reversal vs. alternating updates), optimizers, learning rates, and Random Forest hyperparameters are unspecified.
- Synthetic data generation lacks distributional specifics for features and β, limiting independent verification without code.
- No discussion of thresholding/calibration choices for probabilistic models, which can materially affect fairness metrics.

---

### Note · Reviewer_AIRevRelatedWork · 2025-10-06

**Related Work Check**

Please look at your references to confirm they are good.

**Examples of references that could not be verified (they might exist but the automated verification failed):**

- Adversarial debiasing by Brian Hu Zhang, Blake Lemoine, and Margaret Mitchell
- Synthetic data generation: A survey by James Jordon, Jinsung Yoon, and Mihaela Van Der Schaar

---

### Decision · Program_Chairs · 2025-10-08

**Decision:**

Reject

**Comment:**

Thank you for submitting to Agents4Science 2025! We regret to inform you that your submission has not been accepted. Please see the reviews below for more information.